# A mathematical, classical stratification modeling approach to disentangling the impact of weather on infectious diseases: A case study using spatio-temporally disaggregated *Campylobacter* surveillance data for England and Wales

**Giovanni Lo Iacono**[1,2,3,4]*, Alasdair J. C. Cook[1], Gianne Derks[4,5], Lora E. Fleming[6], Nigel French[7], Emma L. Gillingham[8], Laura C. Gonzalez Villeta[1], Clare Heaviside[9], Roberto M. La Ragione[1,10], Giovanni Leonardi[8,11], Christophe E. Sarran[12], Sotiris Vardoulakis[13], Francis Senyah[14,15], Arnoud H. M. van Vliet[1], Gordon Nichols[1,6,8,16]

**1** Department of Comparative Biomedical Sciences, School of Veterinary Medicine, University of Surrey, Guildford, United Kingdom, **2** Institute for Sustainability, University of Surrey, Guildford, United Kingdom, **3** People-Centred Artificial Intelligence Institute, University of Surrey, Guilford, United Kingdom, **4** Centre for Mathematical and Computational Biology, University of Surrey, Guilford, United Kingdom, **5** Mathematical Institute, Leiden University, Leiden, the Netherlands, **6** European Centre for Environment and Human Health, University of Exeter Medical School, Truro, Cornwall, United Kingdom, **7** New Zealand Food Safety Science & Research Centre, Massey University, Palmerston North, New Zealand, **8** UK Health Security Agency, Chilton, United Kingdom, **9** Institute for Environmental Design and Engineering, University College London, London, United Kingdom, **10** School of Biosciences, University of Surrey, Guilford, United Kingdom, **11** London School of Hygiene and Tropical Medicine, London, United Kingdom, **12** Met Office, Exeter, United Kingdom, **13** Healthy Environments And Lives (HEAL) National Research Network, Australian National University, Canberra, ACT, Australia, **14** UK Health Security Agency, Porton Down, United Kingdom, **15** Médicines Sans Frontièrs, London, United Kingdom, **16** University of East Anglia, Norwich, United Kingdom

* g.loiacono@surrey.ac.uk

## Abstract

Disentangling the impact of the weather on transmission of infectious diseases is crucial for health protection, preparedness and prevention. Because weather factors are co-incidental and partly correlated, we have used geography to separate out the impact of individual weather parameters on other seasonal variables using campylobacteriosis as a case study. Campylobacter infections are found worldwide and are the most common bacterial food-borne disease in developed countries, where they exhibit consistent but country specific seasonality. We developed a novel conditional incidence method, based on classical stratification, exploiting the long term, high-resolution, linkage of approximately one-million campylobacteriosis cases over 20 years in England and Wales with local meteorological datasets from diagnostic laboratory locations. The predicted incidence of campylobacteriosis increased by 1 case per million people for every 5° (Celsius) increase in temperature within the range of 8°−15°. Limited association was observed outside that range. There were strong associations with day-length. Cases tended to increase with relative humidity in the region of 75–80%, while the associations with rainfall and wind-speed were weaker.

**Data Availability Statement:** Dataset and R codes are available at: https://gitlab.surrey.ac.uk/gl0020/campylobacter-linked-with-original-medmi-data.

**Funding:** This research was funded in part by: the UK Medical Research Council (MRC) and the UK Natural Environment Council (NERC) for the MEDMI Project:MR/K019341/1v (LF, GN, CS). The European Union's Horizon 2020 Research and Innovation programme under grant agreement No 773830: One Health European Joint Programme (RMLR, GL,LGV); the National Institute for Health Research Health Protection Research Unit (NIHR HPRU) in Environmental Change and Health at the London School of Hygiene and Tropical Medicine in partnership with Public Health England, and in collaboration with the University of Exeter, University College London, and the Met Office (GN, LF, GL). CH is funded by NERC fellowship (NE/R01440X/1) and Wellcome Trust HEROIC project (216035/Z/19/Z) The funders had no role in study design, data collection and analysis, decision to publish, or preparation of the manuscript.

**Competing interests:** The authors have declared that no competing interests exist.

The approach is able to examine multiple factors and model how complex trends arise, *e.g.* the consistent steep increase in campylobacteriosis in England and Wales in May-June and its spatial variability. This transparent and straightforward approach leads to accurate predictions without relying on regression models and/or postulating specific parameterisations. A key output of the analysis is a thoroughly phenomenological description of the incidence of the disease conditional on specific local weather factors. The study can be crucially important to infer the elusive mechanism of transmission of campylobacteriosis; for instance, by simulating the conditional incidence for a postulated mechanism and compare it with the phenomenological patterns as benchmark. The findings challenge the assumption, commonly made in statistical models, that the transformed mean rate of infection for diseases like campylobacteriosis is a mere additive and combination of the environmental variables.

## Author summary

There is good evidence that weather influences some infectious diseases, driving the seasonal and geographic distribution. This is relevant to gastrointestinal infections, which cause high morbidity and mortality worldwide. Weather can impact people's behaviour, pathogen survival and distribution, animal husbandry, and other environmental variables. We used campylobacteriosis in England and Wales as a case-study to examine a new methodology because it has a distinctive seasonality. The approach compares daily data on affected people by laboratory, the population of the laboratory catchments, and local weather variables. This allows Campylobacter incidence to be compared across the values of these variables (*e.g.* low to high temperature) to provide an estimate of how individual weather variables, alone or in combination, affect disease incidence. We call this the Comparative Conditional Incidence. The results from this analysis are used to build a mathematical model that represents how weather through the year influences the seasonality of disease. The factors most associated with Campylobacter are day-length, air temperature and relative humidity. This will influence future research to understand why the environment influences disease occurrence, and what the burden and pattern of the disease under different climatic scenarios. The methods may have relevance to other seasonal diseases.

## Introduction

Infectious diseases are an important cause of morbidity, mortality, and healthcare and other economic costs worldwide [1]. Despite an encouraging decline in mortality rates in neonates and children under-5 globally [2, 3], especially in south and southeast Asia and South America [4], diarrhoeal diseases were the second leading cause of death in children aged 1–59 months in 2015 [2], and accounted for 9.9% (95% Uncertainty Interval [8.3–11.6]) of deaths in under-5 mortality in 2019 [3]. Diarrhoeal diseases are also a common cause of outpatient visits and hospital admissions in high income countries [5]. The problem is expected to be exacerbated by global population growth, the rising resistance to antibiotics [6] and anthropogenic activities which are constantly changing the environment [7]. The environment, and in particular climate, can affect pathogen abundance, survival, virulence, behavior and host susceptibility to infection as well as human behavior (and vice-versa) [8]. A recent review

found that 277 human pathogenic diseases can be aggravated by climate change, these include 58% of all infectious diseases known to have affected human civilization [9]. Untangling the impact of the environment on infectious diseases is an essential task particularly in the context of climate change. The impact of hydrological and meteorological factors on diarrhoeal diseases is now well documented (see [10] and references therein); however, the exact nature of the associations between meteorological exposures, such as temperature, rainfall and humidity, and the occurrence of the pathogen (*e.g.*, its prevalence) is not fully understood. To this end Colston *et al.* [10] investigated the associations between eight hydro-meteorological factors and a range of enteropathogen. The study reveals the particular influence of temperature and soil moisture across the studied pathogens and highlights the complex and often non-linear associations between the environmental factors and the pathogen prevalence.

Thus, fundamental issues, also relevant for public health, are: i) elucidating how influential environmental factors impact on the disease; ii) exploiting this information to accurately predict the current and future human health risk from infectious diseases; iii) quantifying the time-lag, (*i.e.* the elapsed time during which the specific weather variables contribute to the occurrence of an infection, and not a temporal gap during which the weather has no-effects) between relevant changes in different environmental factors and the emergence of the disease and, iv) gaining a clearer understanding of the transmission dynamics and epidemiology of diseases. Statistical and mathematical approaches can provide promising predictive tools, but are subject to many challenges including: limited knowledge of the mechanism of transmission, collinearity in exposures (*i.e.* highly correlated predictor variables in statistical models), and limitations in available infectious disease data and their linkage with environmental variables [11].

*Campylobacter* is an important example of a climate-sensitive infectious agent. It is one of the most common bacterial foodborne pathogens worldwide [12] and one of the four key global causes of diarrhoeal diseases with rotavirus, typhoid fever and cryptosporidiosis [13]. Moreover, approximately one-third of Guillain-Barré syndrome cases have been attributed to *Campylobacter* infection globally [13]. In 2010 alone, campylobacteriosis was responsible for 166 million [95% Uncertainty Interval 92–301 million] diarrhoeal illnesses, resulting in in 37, 600 deaths worldwide (95% Uncertainty Interval 27, 700–55, 100) [12]. Cases in the UK are known to be underestimated; for every case of campylobacteriosis reported to national surveillance there were 9.3 cases (95% Confidence Interval 6—14.4) in the community [14], and its total UK societal cost is estimated at over £700 million per annum (see [15] and references therein). The origins of *Campylobacter* infection and the routes of transmission are still not fully understood [16]. For instance, current evidence suggest that *Campylobacter* in poultry are responsible for most human campylobacteriosis [15]; still, reported human infections have remained relatively constant despite reduced levels of *Campylobacter* in fresh poultry at retail outlets in the UK [15, 17].

The incidence and prevalence of campylobacteriosis exhibits consistent seasonality, with peaks in May-June in the UK. Potential explanations for such patterns include: seasonality of the incidence in the animal reservoirs; changes in exposure due to human behavior, *e.g.* frequency of barbecues [18]; and seasonality in the abundance of flies which might act as mechanical vectors of the infection [19]. None of these potential causes have conclusively explained and/or predicted the steep increase in incidence during the late spring/early summer [20, 21] observed in England and Wales. Increased temperature has been associated with campylobacteriosis, but this is still rather inconclusive since the pathogen is unable to multiply outside the intestines of warm-blooded animals. There is indication, however, that low temperatures, and protection from the effects of UV and desiccation, favour the survival of

*Campylobacter* [22, 23]. Weather factors (*e.g.* ambient temperature, humidity, rainfall, etc.) are expected to be important drivers [20, 21, 24], but their association is often not certain and the factors are highly-correlated. Furthermore, it is unclear when these factors start to have an effect on the likelihood of infection, and for how long (time-lag).

The aim of this analysis was twofold:

- to provide an in-depth description and quantification of how the weather affects the incidence of campylobacteriosis;

- to develop and validate a predictive approach generalisable to a range of communicable and non-communicable diseases and based on their environmental drivers/factors.

This was done by exploiting long term, high-resolution, epidemiological data linked with local meteorological datasets. The approach estimates the incidence of the disease conditional on specific environmental variables. It is based on classical stratification and relies on a limited set of assumptions.

## Materials and methods

A description of the method is illustrated in Fig 1.

The catchment areas of the diagnostic microbiology laboratories, and the annual population of each catchment, were determined.

Each reported *campylobacter* case was linked to local weather factors (temperature, relative humidity and day-length) as run up to 2010.

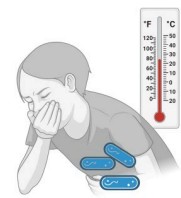

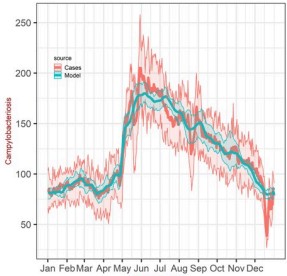

We then estimated the ratio of the number of daily campylobacter cases and the number of residents exposed to the same weather factors (irrespective of date and location) leading to a **conditional probability** or **incidence,** when is expressed as cases per 1,000,000 population per day. The same was done for different lag periods.

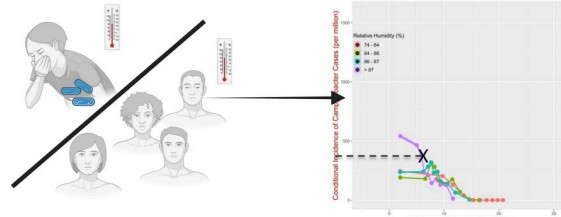

..leading to an estimate of the expected number of cases based on the three weather variables recorded in a particular location at a particular day. The model predictions were plotted against the measured cases in England and Wales.

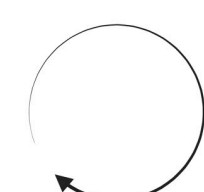

For each catchment and for each day, we ascertained the temperature, relative humidity and day-length (averaged over different lag periods) over 20 years.

Then, we estimate the corresponding conditional probability (marked by X) and multiply by the number of residents..

**Fig 1. Illustrative description of the method. Created with BioRender.com.**

## Data

We used *Campylobacter* data collected by UK Health Security Agency (UKHSA, formerly Public Health England) through national surveillance in England and Wales between 1 January 1989 and 14 April 2010 (Secondary Generation Surveillance System database). *Campylobacter* strains were isolated from stool samples in primary diagnostic laboratories using *Campylobacter* specific agar, and were identified by methods outlined in UKHSA Standard Operating Procedures. The reported temporal data were adjusted from day of year using a 7-day rolling mean, systematic adjustments for the reduced reporting over bank holidays [20] The 994, 791 reported cases were linked to local weather parameters provided by the Met Office statistically interpolated to the locations of diagnostic laboratories *via* the MEDMI platform [25, 26]. Weather variables were available for the same period. Known travel-related cases, (based on the information reported in the form associated with the specimen) were excluded from the analysis. The linkage was conducted on laboratory address rather than patient residential address on the basis that the discrepancies between some weather variables (temperature and rainfall) between the laboratory and the corresponding patient location are known to be small [20, 27].

Daily variables available for the same time period for England and Wales included: maximum and minimum air temperature (degree Celsius °), rainfall (*mm*), relative humidity (%) and mean wind speed (knots). The date when the patient's specimen was collected was not available, therefore data were extracted for the date when specimen reached any of the 416 UKHSA diagnostic laboratories in England and Wales (although the data shows that only 213 laboratories were used for diagnostic, see section 'Regional structure of UK Health Security Agency, diagnostic laboratories and their catchment areas' and Fig A in S1 Text, Fig 3 in [27] and [28]) and for at least any previous 365 days; day-length was calculated from the day of the year, and the latitude of the laboratories [29].

The weather was linked to the day when the infection most likely occurred. As such, the *Campylobacter* specimen dates were adjusted by taking into account a log-normal distributions of the incubation period (mean: 2–5 days [16], Fig B in S1 Text) and a uniform distribution for the reporting delay (range: 1–4 days, based on expert opinion) before infections are finally notified to health authorities [30, 31]. Therefore, for each campylobacteriosis case, we numerically generated a random realization of the incubation period and of the reporting delay drawn from the corresponding distributions. The sum of these two random numbers, $n_{delay}$, corresponds to a random realization of the overall delay since infection. Each campylobacteriosis case was then assumed to occur $n_{delay}$ days before the date when the specimen was received by the diagnostic laboratories (Fig C in S1 Text). We restricted our analysis to cases when the infection likely occurred (*i.e.*, the day after removing incubation period and of the reporting delay) between the 1 January 1990 and 31 December 2009 to ensure complete number of years and that all postcodes were represented during those years. Distribution and correlations of weather variables are shown in Figs D and E in S1 Text. Dataset and R codes are available at https://gitlab.surrey.ac.uk/gl0020/campylobacter-linked-with-original-medmi-data.

## Analysis methods. The Comparative Conditional Incidence

Here we assumed that *Campylobacter* cases in humans are caused by independent, random exposures to a generic source of infection (*e.g.* consumption of contaminated food or contact with contaminated water); while person-to-person transmission is considered negligible. In addition, we assumed that public health interventions have not had any impact on incidence and that human behavior was not changing during the study period. The risk is potentially dependent on a range of factors, but here we focused on the weather variables mentioned

above. Here and throughout we refer to these factors (*i.e.* the potential drivers of the disease) as 'explanatory variables' or 'predictors'.

A common approach to investigating the dependence of disease risk on an explanatory variable in the presence of confounders is by fitting a regression model using a conditional likelihood. Here we adopted the same principle, but we exploited the high spatio-temporal resolution linkage between epidemiological and weather data to ascertain the observed incidence during a certain period, conditional on the weather factors. As a result the method does not rely on parameter fitting or assumptions of the functional form of the dependence of the response on the explanatory variables.

**Mathematical analysis for the incidence of campylobacteriosis conditional on weather factors.**   To simplify the notation, we will present the method for the specific cases when we used three explanatory variables, and these were maximum air temperature $T^{max}$, relative humidity $RH$ and day-length $D$ (day-length is an astronomical factor, but to simplify the language, here we include day-length among the weather factors). The formalism can be readily adapted for rainfall and wind speed too and/or to the situations when we used one, two or four explanatory variables. The corresponding time series evaluated on discrete time values $\Delta t$, $2\Delta t$, ..($\Delta t = 1$ day), and at the laboratory catchment area $\mathbf{x}$, are represented, respectively by $T_{t,\mathbf{x}}^{max}$, $RH_{t,\mathbf{x}}$ and $D_{t,\mathbf{x}}$. The expressions $\overline{T_{t,\mathbf{x}}^{max}}^p$, $\overline{RH_{t,\mathbf{x}}}^p$ and $\overline{D_{t,\mathbf{x}}}^p$ indicate that the said time series have been averaged over the previous time interval $[t - p\Delta t, t]$ and $\overline{T_t^{max}}^p$, $\overline{RH_t}^p$ and $\overline{D_t}^p$ when further averaged over all catchments areas.

It is convenient to discretise each explanatory variable into a finite number of intervals of sizes $\Delta T^{max} = 1°C$, $\Delta RH = 5\%$ and $\Delta D = 1$ hour. Variations within each interval are assumed to have a negligible effect on the risk of campylobacteriosis.

We are interested in the subset of the time series of campylobacteriosis when the values of all explanatory variables, except one, are fixed (in practice within a narrow range), thus we use the notation: $OC(\overline{T_{t,\mathbf{x}}^{max}}^p | \overline{RH_{t,\mathbf{x}}}^p = RH_i, \overline{D_{t,\mathbf{x}}}^p = D_j)$ to represent the number of observed cases at time $t$, in the catchment area $\mathbf{x}$, where the maximum air temperature averaged over the past $p$-days, $\overline{T_{t,\mathbf{x}}^{max}}^p$, can assume any value, but the remaining variables averaged over the past $p$-days, are fixed within the bins $i\Delta RH \leq \overline{RH_{t,\mathbf{x}}}^p \leq (i+1)\Delta RH$ and $j\Delta D \leq \overline{D_{t,\mathbf{x}}}^p \leq (j+1)\Delta D$, these bins are represented by $RH_i$ and $D_j$. If we focus on other weather variables, the notation is modified accordingly. We then define the conditional daily incidence (which can be interpreted as a probability) $P_{Campyl}^p$ as:

$$P_{Campyl}^p(T^{max}|RH_i, D_j) = \sum_t \sum_{\mathbf{x}} \frac{OC(\overline{T_{t,\mathbf{x}}^{max}}^p = T^{max}|\overline{RH_{t,\mathbf{x}}}^p = RH_i, \overline{D_{t,\mathbf{x}}}^p = D_j)}{N(\overline{T_{t,\mathbf{x}}^{max}}^p = T^{max}|\overline{RH_{t,\mathbf{x}}}^p = RH_i, \overline{D_{t,\mathbf{x}}}^p = D_j)} \tag{1}$$

where $N(\overline{T_{t,\mathbf{x}}^{max}}^p | \overline{RH_{i,\mathbf{x}}}^p, \overline{D_{j,\mathbf{x}}}^p)$ is the total number of people in the catchment area $\mathbf{x}$ at time $t$ who have been exposed to the weather variables fixed by the same constraints in the definition of $OC(\overline{T_{t,\mathbf{x}}^{max}}^p | \overline{RH_{t,\mathbf{x}}}^p = RH_i, \overline{D_{j,\mathbf{x}}}^p = D_j)$. In other words, this is the ratio of the number of campylobacteriosis cases given the weather (therefore, this is represented by $OC$) and the total number of people exposed to the same weather conditions. Here and throughout, by conditional incidence $P_{Campyl}^p$, we refer to the daily average number of cases per million people. Accordingly, the notation $P_{Campyl}^p(\overline{T^{max}}|RH_i, D_j) = P_{Campyl}^{14}(20|76, 15) = 2$ means that we expect to observe 2 daily campylobacteriosis in any catchment area where the maximum air temperature, relative humidity and day-length, averaged over the past 14 days, were 20°C, 76% and 15 hours respectively.

The approach presents the following advantages (see also analysis below using Agent Based Models): i) by plotting the conditional incidence $P^p_{Campyl}(\overline{T^{max}}|RH_i, D_j)$ *vs* the explanatory variable $\overline{T^{max p}_t}$, we are able to detect how this is associated with *Campylobacter* infection stratified by all other potential explanatory variables; ii) if $\overline{T^{max p}_t}$ is not a true explanatory variable, the plots of $P^p_{Campyl}(\overline{T^{max}}|RH_i, D_j)$, for all possible stratifications, should show no significant variation (apart random effects) with $\overline{T^{max p}_t}$; iii) if a particular variable, *e.g.* the relative humidity $\overline{RH^p_{t,\mathbf{x}}}$, is a confounder, the plots of $P^p_{Campyl}(\overline{T^{max}}|RH_i, D_j)$ are expected to collapse to the same curve (apart random variations) irrespective of the value of $RH_i$ ($i = 1, 2..$), otherwise (*i.e.* if relative humidity is an effect modifier) the plots of $P^p_{Campyl}(\overline{T^{max p}_t}|\overline{R^p_i}, D_j)$, are expected to result in a series of independent curves for the different values of $RH_i$ ($i = 1, 2..$). In presenting the conditional incidences, the range of the underlying data are divided into quantiles, *e.g.* continuous intervals with equal number of observations, hence the size of these intervals decreases in regions when the frequencies of data is high (compare with Fig D in S1 Text).

**Reconstruction of campylobacteriosis.** In the simplest scenario, the human population in a defined region is uniformly subjected to random and independent exposures to *Campylobacter* infections. Accordingly, occurrences of *Campylobacter* infections in humans are treated as a Poisson process with rate $\lambda(t, \mathbf{x})$ depending on time $t$ and location $\mathbf{x}$, here chosen to be the laboratory catchment (Section A in S1 Text). The expected (reported) number of daily infections is evaluated as the product:

$$\lambda(t, \mathbf{x}) = 10^6 N_{t,\mathbf{x}} P^p_{Campyl}(\overline{T^{max}}|RH_i, D_j) \tag{2}$$

where $N_{t,\mathbf{x}}$, which is the number of people living at the said location $\mathbf{x}$ at time $t$, is approximated as the annual average number of residents in the laboratory catchment, (see Fig A in S1 Text); $P^p_{Campyl}$ is evaluated at the weather factors recorded at location $\mathbf{x}$ at time $t$, the factor $10^6$ is introduced because $P^p_{Campyl}$ is defined per million. Thus, for each catchment area $\mathbf{x}$ at any time $t$ (temporal resolution 1 day), we ascertain the values of the weather factors $\overline{T^{max p}_{t,\mathbf{x}}}$, $\overline{RH^p_{t,\mathbf{x}}}$ and $\overline{D^p_{t,\mathbf{x}}}$, the corresponding conditional incidence $P^p_{Campyl}$ and the number of residents exposed to the same weather variables, in practice $N_{year,\mathbf{x}}$ for each catchment area and year (Fig A in S1 Text).

When we sum over all catchment areas, we obtain the predictions for the expected number of daily infections in England and Wales. The underlying assumption is that the infection rate does not explicitly depend on time and location, the space-time dependence occurs only implicitly *via* the local weather variables experienced during a certain time preceding the infection (but see Section H in S1 Text and corresponding discussion). We also assume that all infections occurring in a particular laboratory catchment are reported to the corresponding laboratory. This expected number of infections, however, is still affected by reporting bias (as the estimation of $P^p_{Campyl}$ was based only on reported cases) although the averaging process over time and locations is expected to reduce heterogeneities in reporting bias.

It is worth noticing that, because of the environmental stochasticity in the weather variables, the rate of infection is itself a stochastic term and thus the underlying Poisson process is strictly a doubly stochastic Poisson process which naturally results in an over-dispersion of the data (*i.e.* a variance larger than the mean) [32]. In the special case that the rate of infection is, or can be approximated by, a gamma-distributed variable, then the process is described by a negative binomial distribution. In the rest of the analysis, however, we focus only the expected rate of infections $\lambda(t, \mathbf{x}))$. The weather variables have been averaged over a fixed, past number of days (14 days unless otherwise specified). Underlying this choice is the assumption that the weather factors continuously contribute to the occurrence of an infection for a fixed time-lag

and that this effect can be captured by their temporal average over the said time-lag. The effect of other time-lags was also assessed in the Supporting Information (Section I in S1 Text).

**Validation with agent based models.** To validate the method, we applied the approach to a synthetic epidemic simulated by an independent Agent Based Model (ABM). The ABM generates random numbers drawn from a Poisson distribution with the rate given by the product of the number of residents in a specific catchment area and a conditional incidence arbitrarily chosen. The time-lag is also arbitrarily chosen (see Section D in S1 Text) and the catchment area was the area associated with the reference laboratory in Welwyn Garden City in England. These random numbers represent the number of new disease cases observed per day. We then apply the method described above assuming no prior knowledge of the conditional incidence and time-lag. The exercise shows that the method correctly identified i) the shape of the conditional incidence (a triangular one in this example), ii) the number of explanatory variables (when relative humidity was not an explanatory variable all the profiles for the conditional incidence collapsed to the same curve as expected) and the time-lag (the reconstruction of the time-series using the incorrect time-lag led clearly showed discrepancies with the time-series generated by the ABM). More details are presented in Section D and Figs F and G in S1 Text.

## Results

### Identifying the more influential weather explanatory factors and how they impact on the disease

We investigated the impact of multiple combinations of weather factors, either simultaneously or separately, on the impact on the disease. Our findings suggests that day-length, maximum air temperature and relative humidity, compared to wind speed and rainfall are the more influential weather factors. The conditional incidence of campylobacteriosis is associated with maximum air temperature (Fig 2A). The conditional incidence tends to be approximately constant for temperatures below 8˚C, this is followed by a sharp increase of about 1 case per million for every 5˚C rise in temperature, where temperatures were between 8˚C and 15˚C and no further rise at temperatures over 15˚C (Fig 2A). The same patterns can be discerned, at least qualitatively, in Fig H in S1 Text, when we stratified the conditional incidence by relative humidity, rainfall, mean wind speed and mean day-length respectively and in Fig 3 when we stratify by relative humidity and mean day-length simultaneously. The conditional incidence associated with minimum air temperature exhibit similar qualitative patterns (Fig 2B, see also the analysis is shown in Fig I in S1 Text, and in Fig J in S1 Text for the difference between maximum and minimum air temperature. As the predictions did not qualitatively change compared to those in Fig H in S1 Text, we continued using maximum air temperature for the rest of the analysis).

The conditional incidence also depends on the relative humidity; this is supported by the observation: i) of a peak around 75–80% in the conditional incidence when this is stratified by relative humidity alone (Fig 2C), ii) that the various profiles for the incidence conditioned to relative humidity are, in general, not collapsing on the same curve (Fig 3) and/or iii) the profiles for the conditional incidence *vs* relative humidity are in general not constant (Fig K and Fig O in S1 Text). The analysis revealed, however, subtle patterns in *how* the conditional incidence specifically depends on relative humidity. For shorter day-length (< 10 hours) relative humidity has limited influence on the conditional incidence. This is shown by all profiles overlapping in Fig 3A but they differentiate in panels B, C and D in Fig 3, and by the fact that the slope of the conditional incidence in Fig K (panel A) in S1 Text is relatively lower compared to the slope of the profiles Fig K (panel C and D) in S1 Text. Similarly, the slope of the profiles for conditional incidence in Fig O for day-length < 9.8 hours tend to be flatter than those for day-length > 9.8, although there are exceptions like the situation when the maximum air

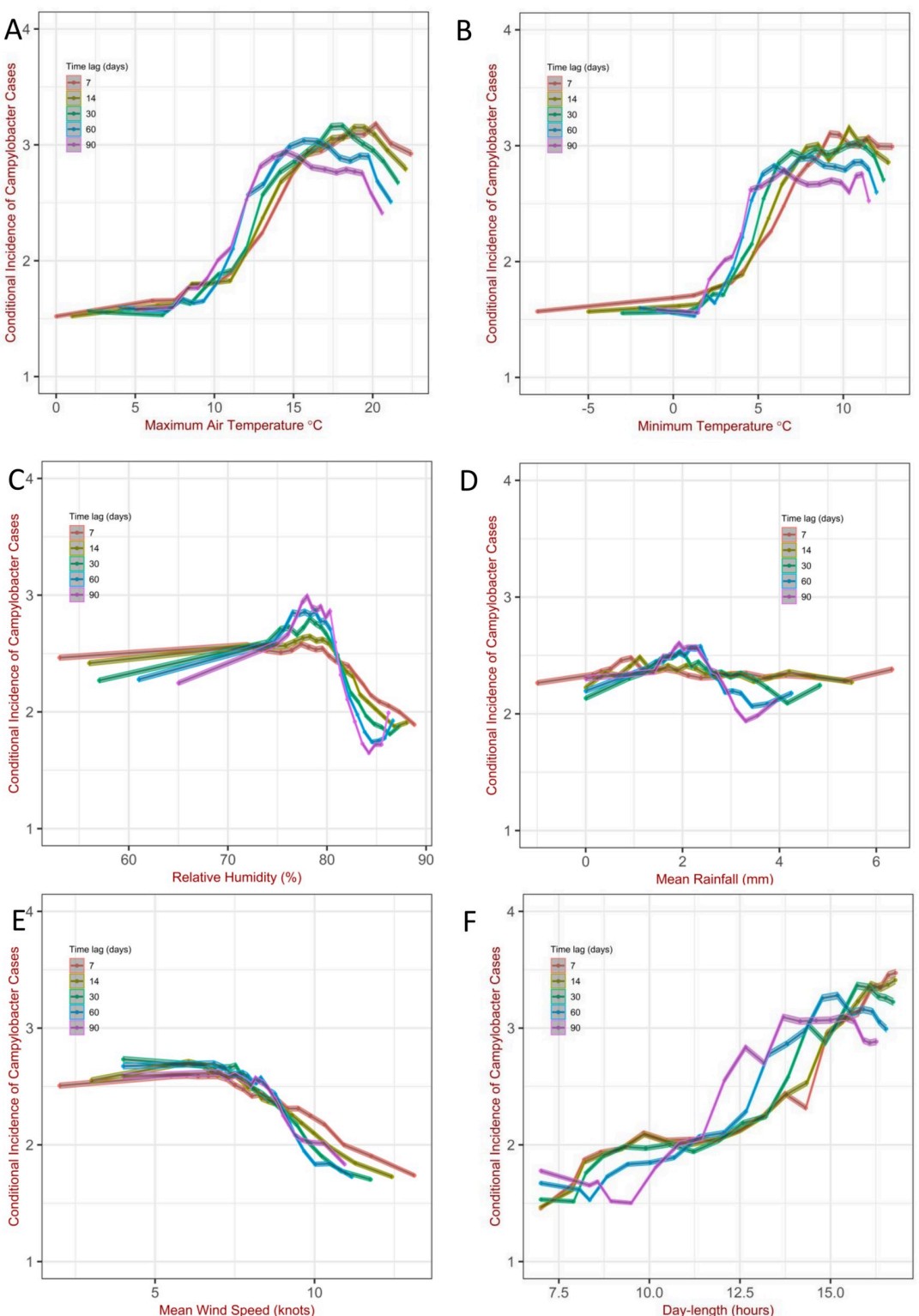

**Fig 2. Campylobacteriosis cases per 1, 000, 000 per day** *vs* **(A) maximum air temperature, (B) minimum air temperature (C) relative humidity (D) rainfall (E) mean wind speed and (F) day-length.** Data were averaged over the past number of days represented by the time-lag. The shaded area shows the 95% confidence intervals for the Poisson means using the normal approximation (*i.e.* average counts $\pm 1.96\sqrt{(\text{average counts}/\text{sample size})}$). Data divided by quantiles.

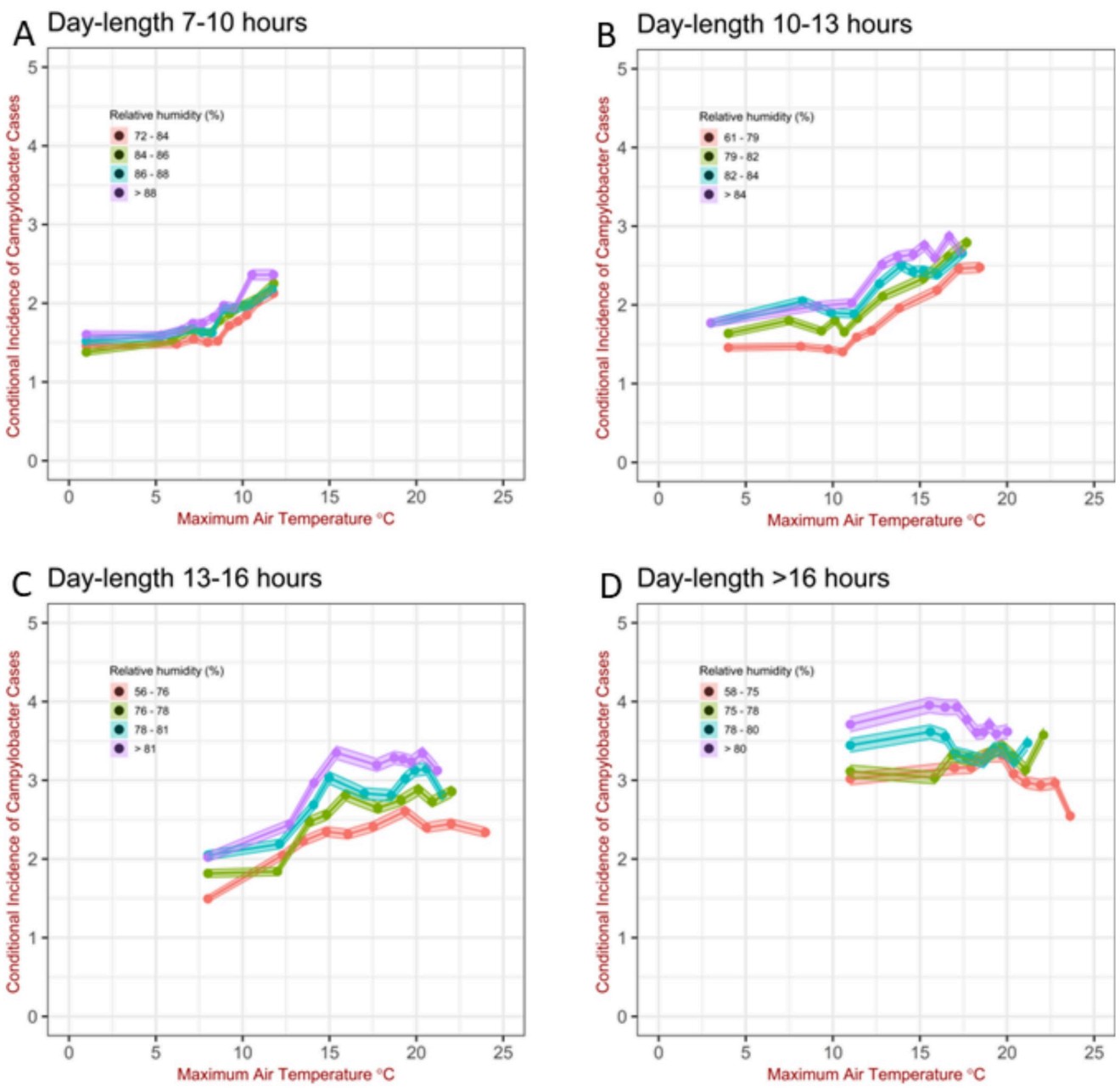

**Fig 3. Campylobacteriosis cases per 1, 000, 000 per day conditioned to maximum air temperature, relative humidity and day-length.** As the day-length depends on the time of the year (as well as latitude), each panel broadly correspond to (A) last week of October—middle of February, (B) middle of February—first week of April and middle of September- last week of October (C) first week of April—second-half of May and second-half of July—middle of September (D) second-half of May—second-half of of July 22. Data were averaged over the past 14 days. The shaded area shows the 95% confidence intervals for the Poisson means using the normal approximation (*i.e.* average counts $\pm 1.96 \sqrt{(\text{average counts}/\text{sample size})}$). Data divided by quantiles.

temperature is in the range 9–10° and the rainfall $> 5mm$. When we increase the number of factors used in the stratification, the peak around 75–80% shown in Fig O, becomes less discernible; however, this feature is preserved, at least in qualitatively, in some situations. Depending on maximum air temperature, day-length and also rainfall, the conditional incidence tends to increase with relative humidity either monotonically or reaching a peak in the

region of high relative humidity followed by a decrease, although the exact slope of the curve and location and magnitude of the peak varies (see Figs K and O in S1 Text). Taking this together, the analysis shows an association of the conditional incidence with relative humidity, especially for day-length longer than 10 hours. The quantitative patterns of the association change according to the specific weather factors.

The dependence of the conditional incidence on rainfall is less pronounced compared to relative humidity (Fig 2D). Furthermore, the profiles of the conditional incidence of Fig L in S1 Text, tend to be flatter compared to that one for relative humidity, indicating limited influence of rainfall on campylobacteriosis. Similarly, in Fig M in S1 Text, the profile of the conditional incidence *vs* mean wind speed shows little variation, implying that mean wind speed has limited impact on campylobacteriosis. The clear patterns in Fig 2 (panel E) in S1 Text are likely to arise from correlations in weather factors, since wind speed tends to be lower for longer day-length and for high maximum temperature (Fig E in S1 Text), which are the conditions when the conditional incidence tend to be high. In contrast, Fig N in S1 Text, along with panel F in Fig 2, suggest a strong association of day-length with the conditional incidence of campylobacteriosis.

The analysis was further refined by investigating the patterns in conditional incidence according to four different weather factors simultaneously, namely: maximum air temperature, relative humidity, rainfall and day-length. As shown in Fig O, the conditional incidence tends to increases with relative humidity with occasional peaks in the profile at larger values of relative humidity ($\gtrapprox 75\%$); this can be seen, for instance, for the situation when the maximum air temperature is in the range of 3–10˚, and day-length in the range 9.8–13.2 hours. The analysis also shows the complex nature of the association of rainfall with campylobacteriosis. In some instances the profiles of the conditional incidence largely overlap for the different levels of rainfall (*e.g.* for maximum air temperature $> 20˚$, and day-length in the range 13.2–15.8 hours, suggesting no association) while in other situations (typically for maximum air temperature $> 17˚$ and day-length $> 15.8$) the different profiles are clearly discernible and show that the conditional incidence of campylobacteriosis is higher when rainfall is lower.

Furthermore, we estimated the conditional incidence of campylobacteriosis *vs* maximum air temperature and relative humidity for two different periods of the year. Namely: the time of the year between i) the shortest (winter solstice) and the longest (summer solstice) duration of day-length and between ii) the longest and the shortest duration of day-length. The choice allows comparisons at different months but ensures that the two parts of the year have the same distribution of day-length. As shown in Fig P in S1 Text, depending on when campylobacteriosis cases occur (from summer to winter solstice or from winter to summer solstice) the relationship between the conditional incidence and maximum air temperature and relative humidity is different. This suggests that additional factors, like time or other environmental factors that correlate with time, might be associated with campylobacteriosis.

## Predicting the patterns of campylobacteriosis

To predict the observed patterns of reported campylobacteriosis, we wanted to identify a minimal set of more influential factors that can be used as inputs in parsimonious models (which are computationally less expensive and rely less on dataset availability). Based on the heuristic considerations above, we retained maximum air temperature, relative humidity and day-length in our subsequent analysis. We can anticipate that this set of factors is sufficient to make accurate predictions of cases and on parsimony ground we did not consider time of the year or any additional factor correlated with time of the year. It is important noticing however, that the patterns in conditional incidence, according to maximum air temperature and relative humidity,

differ for different periods of the year (Fig P in S1 Text). Thus including time of the year as fourth factor is likely to further improve the predictions of campylobacteriosis incidence.

Fig 4 shows the predictions for the corresponding expected daily number of cases reported of *Campylobacter* cases compared with the detected cases per catchment area in England and

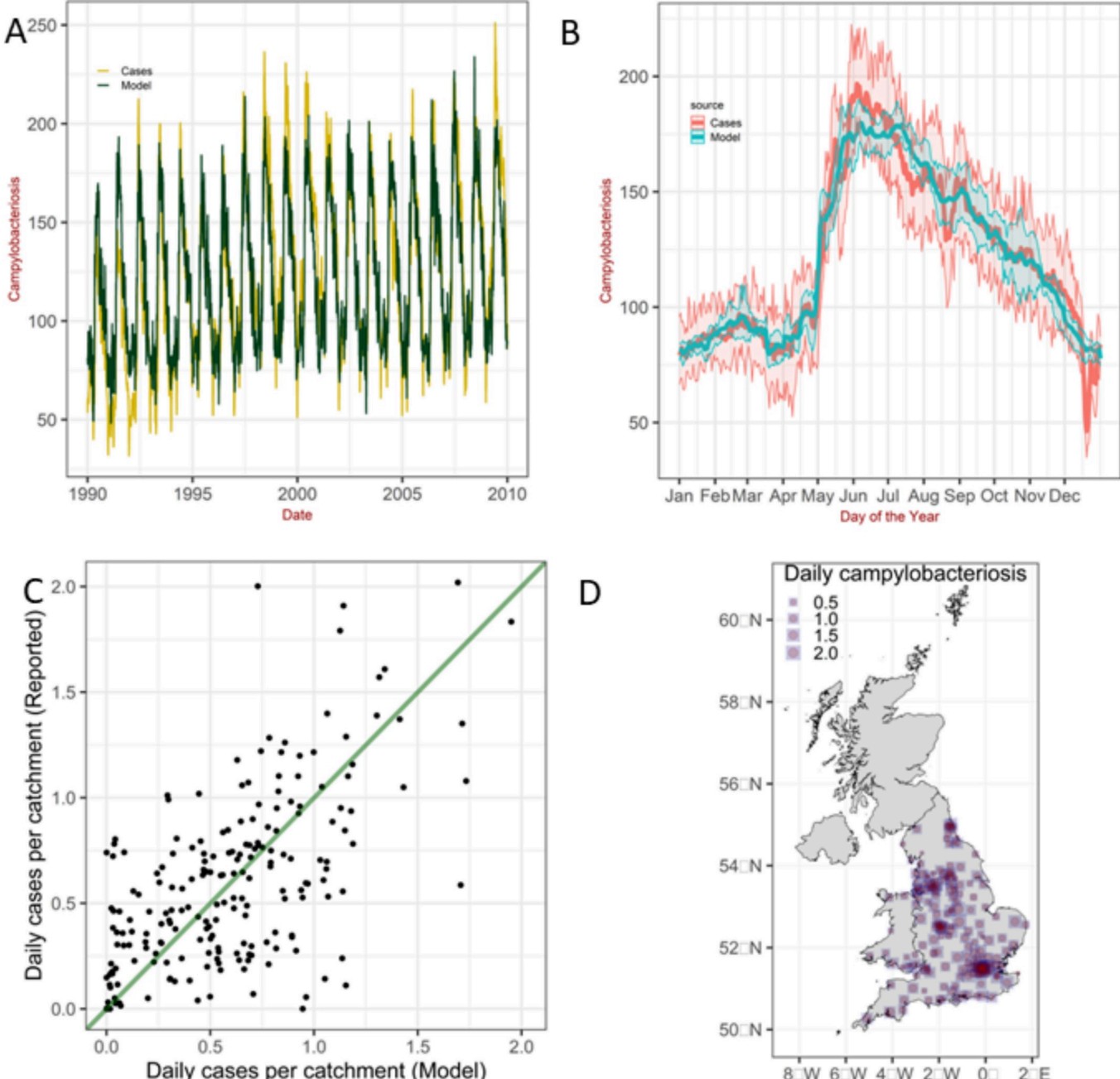

**Fig 4.** A) Reconstruction of the time-series of *Campylobacter* cases in England and Wales. B) Seasonal patterns for daily *Campylobacter* cases averaged over 19 years. The shaded area represents the 25% and *F* quantiles. Weather variables are maximum air temperature, relative humidity and day-length. C-D) Scatter plot and map comparing the reported and predicted daily number of campylobacteriosis per catchment area averaged over the entire 19 years. In D) the red circles represent the reported cases while the blue squares the predictions. Weather variables averaged over the past 14 days. Map reproduced in R [45] using shapefiles availalbe at [46].

Wales (panel A). Panel B shows the corresponding intra-annual variation (seasonality) by aggregating the data monthly. The uncertainty arises from the spatio-temporal variability in temperature, relative humidity and day-length across the different catchments areas and times. The approach also captures the spatial variability as shown in the map and scatter plot in Fig 4C and 4D, which compare the overall reported and predicted daily number of campylobacteriosis per catchment area averaged over the entire 19 years analyzed.

Predictions for the variables averaged over different time-lags (from Fig Q to Fig T, in S1 Text) show only a weak sensitivity of the predictions to different time-lags considered here.

The analysis also showed that the inclusion of day-length greatly enhanced the predictions compared to prediction based on two weather factors alone. To investigate this aspect further, we simulated the time-series of cases first using only one explanatory variable and then two explanatory variables. Predictions using day-length and maximum air temperature separately broadly capture the seasonality in campylobacteriosis, although the model exhibits large discrepancies with the empirical data (Fig U in S1 Text; panels A-B and C-D). Seasonal patterns in the predictions are still discernible when we used relative humidity, while seasonal variation in the predictions are negligible when we used wind speed and rainfall only. Using simultaneously day-length and an additional weather factor improve the accuracy of the predictions, especially for the combination day-length and maximum air temperature (Fig V in S1 Text; panels A-B). Other combinations lead to poor predictions (Fig V in S1 Text). A visual assessment of the predictions of the model combining day-length and maximum air temperature indicates that the predictions for the intra-annual variability of campylobacteriosis are comparable to that when we use three weather factors (compare panels B in Fig 4 and Fig V in S1 Text); this combination, however, results in poor predictions of the inter-annual variability (compare panels A in Fig 4 and Fig V in S1 Text). Taken together, the findings suggest that campylobacteriosis is strongly associated with day-length and maximum air temperature and these two weather factors are sufficient to predict the seasonality of the disease, while a weaker association with relative humidity might be the driver of the inter-annual variability. Thus, even if rainfall alone is a poor predictor, when combined with day-length and maximum air temperature results in accurate predictions as shown in Fig W in S1 Text.

### Impact of individual variables on the seasonality of campylobacteriosis cases

The impact of a single seasonally-varying, specific weather factor on the patterns of campylobacteriosis cases when the other two variables are kept constant is demonstrated in Fig 5 showing that the specific seasonal patterns observed in England and Wales originate from the combination of all three variables and the non-linear functional forms of the conditional incidence. For instance, for a relative humidity of 76% and a maximum air temperature of 20°C the conditional incidence of campylobacteriosis exhibit a maximum in July and a local minimum in May (dashed line in Fig 5C); in these months the day-length is about 17 and 15 hours respectively (Fig 5F) for which the conditional incidence exhibits a local maximum and a local minimum (Fig 5I). These patterns strongly depends on the specific values of the chosen weather factors, the purpose of the analysis of these hypothetical scenarios is to elucidate how the specific seasonal patterns observed in England and Wales originate from the combination of all three variables and the non-linear functional forms of the conditional incidence.

### Discussion

Seasonality is a recognized driver of the temporal patterns of campylobacteriosis as shown in Fig C in the Supporting Information (S1 Text). To understand these seasonal patterns, we

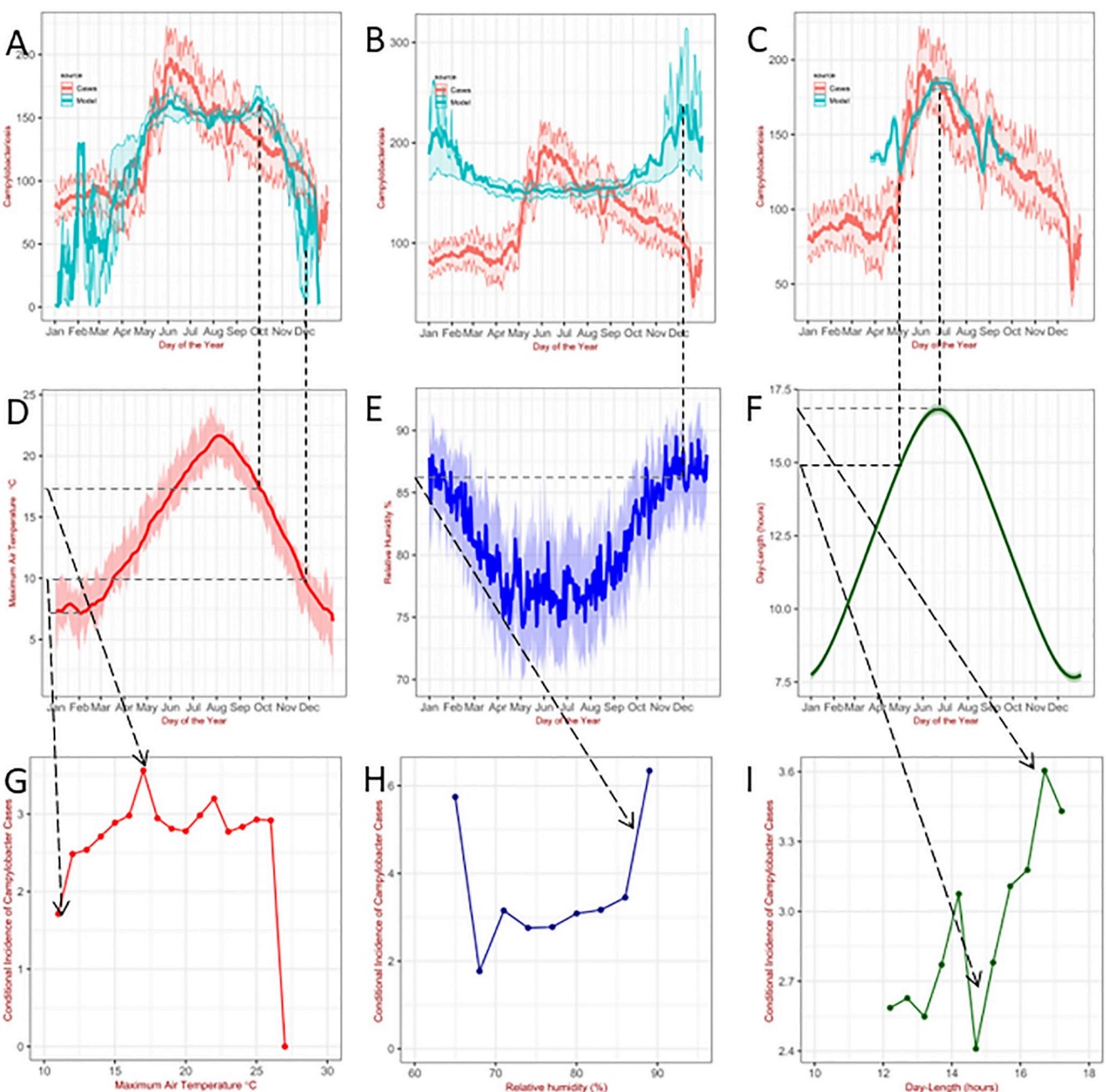

**Fig 5. Prediction of seasonal patterns for daily *Campylobacter* cases as done in Fig 4 for the situation when 2 variables are constant (Weather variables averaged over the past 14 days).** A) Constant relative humidity 76% and day-length 15 hours. B) Constant maximum air temperature 20˚$C$ and day-length 15 hours. C) Constant maximum air temperature 20˚$C$ and relative humidity 76%. D-E-F) Patterns for daily 14-days rolling mean for maximum air temperature, relative humidity and day-length averaged over 19 years. The shaded area represents the 25% and 75% quantiles. G-H-I) Conditional incidence *vs* the variable weather factors for the situation corresponding to A) B) and C) respectively.

collected and linked long term high-resolution epidemiological and weather data allowing for the stratification of the conditional incidence of the disease conditional on specific weather factors. We then systematically investigated the relationship between the conditional incidence of campylobacteriosis and a range of variables: air temperatures, relative humidity, rainfall,

wind speed, day-length and different period of the year for different time-lags. We began by exploring this relationship considering only one weather variable at a time (Fig 2). Despite the clearly detectable patterns, this analysis could lead to incorrect conclusions due to the correlations among some of the variables or simply the high frequency of specific values (Figs D and E in S1 Text). Thus we refined our analysis by stratifying the conditional incidence according to exposure to two (from Fig H to Fig J in Section E in S1 Text), three (Fig 3 and from Fig K to Fig N in Section F in S1 Text) and four specific weather variables (Fig O in Section G in S1 Text) as well as for different period of the year with the the same distribution of day-length (from winter to summer solstice and viceversa) (Fig P in Section H in S1 Text).

The findings show that the conditional incidence of campylobacteriosis was associated with maximum air temperature in a non-linear fashion. For temperatures below 8°C the conditional incidence exhibits no or limited increase, this is followed by about 1 case per million for every 5°C rise in temperature where temperatures were between 8°C and 15°C and no further rise at temperatures over 15°C. The patterns simultaneously depends on relative humidity and day-length. The conditional incidence tend to increase with day-length. Cases had marked higher conditional incidence when the relative humidity was in the region of 75–85% (depending on which factors is used for the stratification). These findings are in line with the observation that the survival of *Campylobacter* in the environment is enhanced by moist conditions [23]; and also with the findings of Kalupahana *et al.* [33], who observed a decrease in *Campylobacter* prevalence in broilers (chickens) at slaughter at high relative humidity ($> 80\%$). Colston and colleagues [10], however, found limited association between incidence of campylobacteriosis and relative humidity. It is important to highlight that their study focused on enteric pathogens in stool samples collected from children aged under 5 years in Low- and Middle-Income Countries, while here we considered all ages in a High-Income country. Another important difference is that Colston *et al.*'s study comprised all exposures and covariate variables including soil moisture and relative humidity. In our work, data on soil moisture were not available and thus the observed association with relative humidity might be arise from correlation with soil moisture and precipitation [34].

In general, associations with rainfall and wind-speed are weak. The fly hypothesis, *i.e.*, that campylobacteriosis is mechanically transmitted by flies (*musca domestica*) [19], appears to contradict the weak association of conditional incidence with wind-speed since flies activity is expected to depend on wind speed. It is worth noticing that ambient temperature has a direct influence on the physical activity and the life span of houseflies and their flight activity increases with temperature [35]. Further research is needed to test this hypothesis.

The analysis shows that the relevant meteorological variables considered here contribute to the conditional incidence in a non-monotonic and non-additive fashion. This is an important result *per se*, as many regression methods need to make precise assumptions on the functional relationships between the predictors and the response variable; for instance, assuming that the rate of infection is a linear combination of the weather variables. The problem could be avoided by using Machine Learning approaches, such as Random Forest which uses an algorithm to learn the relationship between the response and its predictors. A further advancement, boosted regression trees, draws on insights and techniques from both statistics and Machine Learning [36]. These approaches rely on an algorithm processing the data (*e.g.* employing splitting criteria and classifying the predicted class as the most common class in the node). Here we used a classical stratification technique which is not based on any inference or decision, and we argue that the resulting conditional incidence can be interpreted as a gold standard as purely based on a description of the data. Most importantly, compared to Machine Learning approaches, our method and the findings are readily interpretable by non-experts.

Predicting and understanding seasonality and complex trends in the dynamics of infectious diseases is a crucial goal for public health. Environmental data have a great potential to improve the predictive power of models and to understand complex patterns in the spread of diseases. The approach elucidates the seasonal patterns in campylobacteriosis and it is able to predict complex trends, *e.g.* the steep increase in campylobacteriosis in England and Wales in May-June and their spatial variability. It combines information on the conditional incidence with local information on maximum air temperature, relative humidity and day-length without relying on regression models and/or postulating specific parameterisations.

The observed seasonal incidence of *Campylobacter* depend on the particular geographic region under investigation [37], as well as on the particular reporting system employed in the country. These seasonal patterns arise from the non-linear functional forms of the conditional incidences combined with the weather variables observed in England and Wales. Thus rather than asking why there is a steep increase in campylobacteriosis in England and Wales in May-June we should focus on explaining what is the mechanism resulting in the specific shape of the observed incidence conditional on the weather variable. At the moment, we cannot establish if this relationship is universal, or typical of only the situation in England and Wales (implying that it is affected by human behavior, such as higher frequency of barbecues in warmer days, or both). According to a recent survey of *campylobacter* contamination in fresh, whole UK-produced chilled chickens at retail sale, the percentage of samples with more than 1000 cfu of *campylobacter* spp. per gram was significantly higher in the period May, June, and July than in the period November to April [38]. This indicates that the seasonal patterns of campylobacteriosis might be related to the biophysical process, such as survival of the bacteria in response to the environment or food chain rather than individual consumer behavior. In future, it would be interesting to apply the method, using the same conditional incidence for England and Wales, to other geographic settings for example, European countries and southern hemisphere countries (*e.g.* Australia and New Zealand where appropriate datasets are available) and test if the model predicts the correct shift in the time when *Campylobacter* incidence peaks [37]; and if the model captures the different seasonal patterns in New Zealand's North and South Islands, with higher amplitude of peaks in Canterbury and less detectable seasonal pattern in Auckland [39]. Similarly, after including socio-economic factors such as livestock density, the model could be used to investigate the difference in the seasonal pattern in urban versus rural areas (as observed in New Zealand) with peaks during early spring in areas with high cattle density likely to be ruminant-associated, compared to summer in urban areas likely to be attributable to poultry [40].

Agent based models have been used to investigate the elusive mechanism of transmission [41]. Our approach can assist this class of models by comparing, for example, the empirical conditional incidence found here with a simulated incidence conditional to weather factors, based on the hypothesized mechanism.

In this work, we considered only a limited number of weather variables. The approach can be improved by exploring the impact of other variables potentially involved in the causal pathways, such as: dew point temperature, soil temperature, UV radiation (rather than day-length), atmospheric pressure, etc. These data can now be extracted from data collected from meteorological agencies (*e.g.* the Met Office [25, 26]) although they might be subjected to some limitations (*e.g.* temporal and spatial gaps in the data collection and computational resources). Including a higher number of variables is possible, the practical disadvantage, however, is the reduction in the sample size for a particular set of conditions, and the associated greater uncertainty. Nevertheless, other environmental and socio-economic variables can be readily incorporated in this approach. For instance, the inclusion of land use (*e.g.* spreading composted or

creating middens of un-composted poultry litter on land), built up cover, and the spatial distribution of poultry and other livestock, can assist in detecting the impact of urban *vs* rural factors in the incidence of the disease.

## Limitations

A limitation of this study was the use of laboratory location rather than patients' address. The choice, however, presents some advantages: i) the laboratory address database was more complete than the patient residential address since the surveillance system had limited patient postcode records before 2008, ii) the computational resources needed for data linkage at the individual patient postcode level can be significant large, iii) the use of laboratory address prevents challenges to comply with ethical and legal requirements of confidentiality. Furthermore, using the patient's residential address could introduce further bias as it is estimated that people spend up to 40% of time at locations other than their residence address (schools, workplace, transport) [27].

The uncertainty associated with the conditional incidence was quantified by estimating 95% confidence intervals as Wald interval. Other measures could be considered. For instance, the relative risk conditioned to specific weather factors can be estimated by dichotomizing the continuous weather factors at an adequate threshold (*e.g.*, 8˚ for maximum air temperature) and calculating the ratio of the probability of campylobacteriosis for when the weather factor is above the chosen threshold (the "exposed group") to the probability of campylobacteriosis for when the weather factor is below the threshold (the "unexposed group") [10].

The current analysis cannot distinguish if and how day-length is involved in the causal pathways; day-length could be a proxy of the effect of radiation on the survival of the bacterium or a proxy for human behavior driven by the seasonal cycle of day-length. In future research, the data should be stratified simultaneously by day-length and a measure of radiation, *e.g.* UV, and ascertain the 'true' explanatory variable.

An additional limitation of this study is that the data comes from a high-income country, with specific climatic disease-burden, therefore, it may not be possible to extrapolate these findings to the low-resource, tropical settings. An important objective for future research it to asses if the functional form of the conditional incidence depends on specific geographic and socio-economic settings. This was done for a similar study for salmonellosis [42], which showed that the approach accurately reproduces the empirical patterns of salmonellosis in The Netherlands by using the conditional incidence derived from England and Wales data. If the nature of the conditional incidence is proven to be general, then the method can be readily applied to investigate how the burden and patterns of diseases will change due to climate and other environmental changes.

We applied the model to *Campylobacter*, mainly because of the abundance of data. This approach, however, can be readily applied to other diseases for which the temporal dynamics, for instance arising from person-to-person transmission, are negligible or can be incorporated in the model by using time-varying variables (such as day-length) as proxies. It is worth exploring if the approach can be applied to model non-communicable diseases such as cancer due to long exposure to factors such as contaminated air.

An extension of the approach is the inclusion of person-to-person transmission, which would be relevant to infectious gastro-intestinal diseases such as those caused by norovirus, rotavirus and respiratory diseases such as those caused by influenza A viruses and coronaviruses. This can be done by employing Poisson processes with memory of past events, Hawkes processes, and by explicitly separating the contributions arising from the environment from the one from person-to-person [43, 44]. This will be explored in future works.

## Conclusion

We formulated a novel mathematical approach and used a unique dataset which combines a large high-resolution spatio-temporal epidemiological data for campylobacteriosis from UK Health Security Agency, locally linked with multiple environmental factors at high geographic resolution from the Met Office. This leads to a thoroughly description and quantification of the comparative incidence of campylobacteriosis that is conditional to the local weather factors. The analysis controls for geography and time by providing an average of daily case incidence in all catchments, and for the entire time period, measured for individual weather values. The Comparative Conditional Incidence approach also allows simultaneous analysis of two or three weather parameters.

Campylobacter infection is strongly associated, in a complex and non-linear way, with day-length and maximum air temperature and, to a lesser extent, with relative humidity. These three weather factors are sufficient to accurately predict the incidence of the disease. Although method cannot ascertain if the specific weather factors are involved in the causal pathways, the the Comparative Conditional Incidence can provide an insight into how weather factors might affect the mechanism of transmission. This information can be used as benchmark for future agent-based models aiming to investigate the underlying causes of campylobacteriosis transmission.

The Comparative Conditional Incidence approach is generalisable to other environmental-driven communicable and non-communicable diseases and can be applied to investigate how the seasonal burden of diseases will change under different climatic scenarios.

## Supporting information

**S1 Text. The file contains the following section: Regional structure of UK Health Security Agency, diagnostic laboratories and their catchment areas.** Removing Reporting Delays and the effect of Incubation Period. Correlations among the weather variables and their distributions. Validation with Agent Based Models. Patterns in conditional incidence according to different weather variables (two weather factors simultaneously). Different Ways to Visualize conditional incidence (three weather factors simultaneously). Patterns in conditional incidence according to different weather variables (four weather factors simultaneously). Patterns in conditional incidence according to maximum air temperature and relative humidity for different periods of the year. Incidence of campylobacteriosis cases when the weather variables are averaged over different time-lags (three weather factors simultaneously). Seasonal patterns for daily *Campylobacter* cases using only one predictor. Seasonal patterns for daily *Campylobacter* cases using only two predictors. Predictions using rainfall, instead of relative humidity, as predictor.
(PDF)

## Acknowledgments

We would like to thank Prof Jan Semenza for his valuable feedback and comments.

## Author Contributions

**Conceptualization:** Giovanni Lo Iacono, Arnoud H. M. van Vliet, Gordon Nichols.

**Data curation:** Giovanni Lo Iacono.

**Formal analysis:** Giovanni Lo Iacono.

**Funding acquisition:** Lora E. Fleming, Roberto M. La Ragione, Sotiris Vardoulakis, Gordon Nichols.

**Investigation:** Giovanni Lo Iacono, Gordon Nichols.

**Methodology:** Giovanni Lo Iacono, Gordon Nichols.

**Project administration:** Giovanni Lo Iacono.

**Resources:** Giovanni Lo Iacono, Christophe E. Sarran, Francis Senyah, Gordon Nichols.

**Software:** Giovanni Lo Iacono.

**Validation:** Giovanni Lo Iacono.

**Visualization:** Giovanni Lo Iacono.

**Writing – original draft:** Giovanni Lo Iacono.

**Writing – review & editing:** Giovanni Lo Iacono, Alasdair J. C. Cook, Gianne Derks, Lora E. Fleming, Nigel French, Emma L. Gillingham, Laura C. Gonzalez Villeta, Clare Heaviside, Roberto M. La Ragione, Giovanni Leonardi, Christophe E. Sarran, Sotiris Vardoulakis, Arnoud H. M. van Vliet, Gordon Nichols.

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
