## [Decision Letter · Decision Letter 0]

26 Jul 2023

Dear Dr. Lo Iacono,

Thank you very much for submitting your manuscript "Unravelling the impact of weather on infectious diseases: A Campylobacter case study" for consideration at PLOS Computational Biology. As with all papers reviewed by the journal, your manuscript was reviewed by members of the editorial board and by several independent reviewers. The reviewers appreciated the attention to an important topic. Based on the reviews, we are likely to accept this manuscript for publication, providing that you modify the manuscript according to the review recommendations.

Reviewer comments were overall positive and mainly indicated areas where the clarity of the manuscript should be improved.

Sincerely,

Nic Vega, Ph.D.

Academic Editor

PLOS Computational Biology

Thomas Leitner

Section Editor

PLOS Computational Biology

Reviewer comments were overall positive and mainly indicated areas where the clarity of the manuscript should be improved.

Reviewer's Responses to Questions

**Comments to the Authors:**

Reviewer #1: uploaded as an attachment

The study aims to establish associations between epidemiological and weather data using the proposed metric, conditional incidence, which is thoroughly explained by the authors. This research provides valuable scientific evidence for understanding the mechanism of Campylobacter infections.

One important consideration is whether the uncertainty associated with the conditional incidence can be quantified, similar to commonly used measures of association such as Pearson's correlation coefficient, odds ratio, or risk ratio. By estimating the uncertainty linked to the conditional incidence, it would greatly benefit future computational health research studies in assessing the impact of various environmental factors on disease cases.

A minor correction is required in line number 196, where "reconstructed the time-series" should be corrected to "reconstructed time series."

Reviewer #2: This paper describes an innovative non-regression-based method to estimate the association between weather and risk of Campylobacter infection. The authors first investigated the most influential factors on the disease risk then performed a prediction analysis to compare with the observed incidences and seasonality. I enjoyed reading the paper. However, significant revisions are also needed before publication.

Methods

1. Line 66. How did the authors define and exclude travel-related cases?

2. Line 67. The use of laboratory address was still a limitation of this study.

3. Now that the incubation period is 2-5 days and there some 1-4 days for reporting delay, why didn't the author use 7 days as the primary lag time for analysis?

4. Lines 117-119. There indeed are other parameters (rainfall & wind speed) so they need to be stated in Methods.

5. Last paragraph on page 4/29. D(j,x) should be D(t,x)?

6. Page 5/29, all P(p,Campyl) should be Pr(p,Campyl) to be consistent.

7. Please explain OC in equation (1).

8. Lines 140-143. I'm not following the rationale behind. As far as I can understand, if RH is an effect modifier, the plots of Pr(p,Campyl) should be a series of independent curves.

9. Lines 144-147, I'm also confused by the description. Can the authors give a very brief example?

10. "The expected number of daily infections is given by...see Fig S-3": I don't think this is relevant to the study since the authors didn't use R(inf) at all. Just explain the N(year,x) in the following equation.

11. Line 175. I don't any specification on time lags in S1.

12. Line 178. Any reference for the equations? Why 15, 41, and 13 degrees? More importantly, results from ABM are synthetic disease data, as the authors have implied. Then how can this be used for validation of the methods the authors proposed? This directly lead to the question of validity of this approach. The authors should carefully justify it.

13. Lines 187-188. Both figures in Fig 2 are ABM results, not from conditional incidence method.

14. Lines 195-198. As mentioned earlier, estimates from ABM are not real data either.

Results:

15. Line 203. There is no assessment in terms of rainfall and wind speed in Methods.

16. Lines 213-214. I don't think the association is apparent for the mainly focused time lag (14 days) of the entire manuscript . It only becomes obvious when the lag is over 30 days. Also, the authors should also note the negative association between 80-90% which should be discussed in the text. This is much more consistent and apparent than the positive association the authors mentioned for 75-80%.

17. Lines 225-226. I think the authors should explore more on this negative association which seems consistent across time lags, rather than speculating it as a spurious relationship.

18. Line 226. Should also mention Fig 4F.

19. Fig S15 is missing in the result text. The author should re-organize the text to make sure all figures appear in order in the text.

20. Lines 238-239. An analysis similar to what? Need to re-organize the result.

21. Lines 247-248. Why not? The authors have already found significant variations across time , not like rainfall and wind speed.

22. Lines 263-264. The authors should discuss why in conditional incidence approach rainfall seems not to be associated with the cases but in prediction it serves as a necessary component? Or is it because that actually only day-length (with or without maximum temperature) is the most essential element predicting the incidence? This can be easily tested.

23. Lines 265-279. I'm not following. I understand that if only one variable was used with the other two constant (Fig 7A, B, and C), they can not capture the variation of the seasonality. But how can these three individual analysis elucidate that the estimates from Fig 6B are the results originated from the combination of all three variables?

Discussion:

24. Lines 288-296. This should be explicitly somewhere in Methods and Results for the readers to better understand the results and figures (and the motivation behind them).

25. Lines 336-338. Can the authors elaborate more on the comparison between this approach and case crossover? Since it's not a regression-based approach, how are other baseline characteristics be controlled in the analysis?

26. Limitations should be discussed. And there should be conclusions.

The authors should also check thoroughly for grammatic errors.

Reviewer #3: Congratulations on an excellent analysis that is well conceptualized and executed, mostly well written and with appealing and interpretable data visualizations. As the authors convincingly point out, given the exacerbating effects of climate change on infectious disease transmission, it is important to tease out the effects of specific environmental parameters on individual pathogens and the pathways through which these associations operate. To do this requires highly spatiotemporally disaggregated data that is representative of a broad range of climatic contexts, as well as sophisticated statistical methods. In this regard, the manuscript represents a significant advance in addressing the knowledge gaps surrounding climate impacts on a high-burden, vaccine-targeted, diarrhea-causing pathogen, namely Campylobacter. The authors have assembled a rare jewel of a dataset: a repository of some 100,000 Campylobacter diagnostic results from across England and Wales that have been matched spatiotemporally with biologically relevant weather variables while giving appropriate consideration to incubation period and reporting delays. They have also provided the data and code via a GitLab repository. While I am not qualified to comment on the specifics of the mathematical modeling approach used, I do have some more general comments and suggestions that I think need to be addressed in order to make this suitable for publication in PLOS Computational Biology.

Title

I suggest rewording the title to better reflect the aim and design of the study and the data sources used. For example: “A mathematical, classical stratification modeling approach to disentangling the impact of weather on infectious diseases: a case study using spatiotemporally disaggregated Campylobacter surveillance data for England and Wales.”

Abstract

I suggest simplifying the take-away statement to “The predicted incidence of campylobacteriosis increased by 1 case per million for every 5 degree increase in temperature within the range of 8 – 15 degrees. No association was observed outside that range” and making a similar statement for humidity which quantifies the effect size found by the model.

Background

I’m not sure that the authors are always citing the most relevant literature. For the statement about the global burden of diarrheal disease (#2) consider citing https://pubmed.ncbi.nlm.nih.gov/34416195/ and/or https://pubmed.ncbi.nlm.nih.gov/27839855/. For the statement about environmental impact on pathogens (#6) consider citing https://pubmed.ncbi.nlm.nih.gov/35968032/ . In the second paragraph, consider citing https://pubmed.ncbi.nlm.nih.gov/35024531/, a study which tackles most of the fundamental issues that the authors highlight.

I am not familiar with the term “phenomenological” used in this context and I found it a little jarring, though this may be a technical use of the term with which I am not familiar.

Please state the aim of the paper more clearly. Is it to quantify the effects of weather on Campylobacter, or is it to showcase a methodology that can be adapted to other diseases and contexts? There should be a sentence that starts “The aim of this analysis is to…”

Materials and Methods

Please state the units of weather variables. Were total daily rainfall volumes used or averages?

I assume that campylobacteriosis was diagnosed in stool samples, but this is not stated anywhere. What diagnostic method was used (culture? PCR?), and was it standard across laboratories?

The authors explain that cases were georeferenced to the laboratory locations, and not residences due to completeness considerations. However, they don’t explain why they used the date that the samples arrived at the laboratory and not the date when they were collected from the patient. Was this information similarly incomplete? If sample collection date could be used, it would eliminate the need to incorporate reporting delays.

I do not understand what is meant by “We restricted our analysis to cases when the infection likely occurred … between the 1 January 1990 and 31 December 2009 to ensure … that all postcodes were represented.”

Instead of generating a random realization of the incubation period and reporting delay, consider aggregating variables over a lagged window of exposure (see https://pubmed.ncbi.nlm.nih.gov/36796984/).

The justification for including day length as an explanatory variable is not clear to me. As is apparent in figure 7F, this variable has a very smooth, sinusoidal distribution. It strikes me that the effect of including it will therefore be similar to that of including a harmonic term/ Fourier function (https://pubmed.ncbi.nlm.nih.gov/10396550/), i.e., an adjustment for laboratory catchment area-specific seasonality patterns so that the effects of the other variables can be interpreted as short-term, interdiurnal associations (see https://pubmed.ncbi.nlm.nih.gov/31229000/). If this is the case, then this should be stated explicitly, along with a justification for only using just one term, when the seasonal pattern of Campylobacter incidence is irregular (fig S-5). If this is not the case, and the authors instead hypothesize some direct association between day length and the infection outcome (such as the cumulative effect of solar radiation either on the bacteria itself or on host immunity), then this too should be stated. Either way, it is unsurprising that this variable should show an association with disease incidence, since it is essentially a proxy for the seasonal cycle.

Results

Please narrow the range of the axes in Figure 5 so that the lines occupy more of the plot space and the shape of the associations will be more discernable.

Discussion

Compare the results with Colston and colleagues (https://pubmed.ncbi.nlm.nih.gov/35024531/) who likewise found a strong, non-linear association of Campylobacter with temperature - though with an upper plateau at a much higher temperature – and negligible effects of wind speed and rainfall, but did not find an association with relative humidity in a model that adjusted for soil moisture.

I am confused by the comment that this analysis resembles a case-crossover study design.

An additional limitation of this study is that the data comes from a high-income, temperature latitude and low Campylobacter-burden context and therefore, it may not be possible to extrapolate these findings to the low-resource, tropical settings where incidence of the disease and its sequelae are highest.

General comments

I do not fully understand the ABM part of the analysis. I think the justification for this needs to be made more explicit but, since it seems to be some sort of validation exercise, why are the results presented so early on (figs 2 and 3)? As a reader, I want to see the actual results of the model first before seeing proof of its validity, which is of secondary interest. Can Figures 2 and 3 not be relegated to the supplementary materials? There are already a lot of figures in the main article.

I suggest keeping the background, methods, results, and discussion more distinct throughout. For example, the first few sentences of the “analysis methods” section reads like background and cites background literature.

**Have the authors made all data and (if applicable) computational code underlying the findings in their manuscript fully available?**

Reviewer #1: Yes

Reviewer #2: Yes

Reviewer #3: Yes

PLOS authors have the option to publish the peer review history of their article (what does this mean?). If published, this will include your full peer review and any attached files.

Reviewer #1: No

Reviewer #2: No

Reviewer #3: No

Figure Files:

Data Requirements:

Reproducibility:

References:

---

## [Decision Letter · Decision Letter 1]

27 Nov 2023

Dear Dr. Lo Iacono,

We are pleased to inform you that your manuscript 'A mathematical, classical stratification modeling approach to disentangling the impact of weather on infectious diseases: a case study using spatio-temporally disaggregated Campylobacter surveillance data for  England and Wales' has been provisionally accepted for publication in PLOS Computational Biology.

Best regards,

Nic Vega, Ph.D.

Academic Editor

PLOS Computational Biology

Thomas Leitner

Section Editor

PLOS Computational Biology

Reviewer's Responses to Questions

**Comments to the Authors:**

Reviewer #3: THank you for addressing my comments satisfactorily and I hope this article is well received once published (As it deserves to be).

**Have the authors made all data and (if applicable) computational code underlying the findings in their manuscript fully available?**

Reviewer #3: Yes

PLOS authors have the option to publish the peer review history of their article (what does this mean?). If published, this will include your full peer review and any attached files.

Reviewer #3: No

---

## [Editor Report · Acceptance letter]

12 Dec 2023

PCOMPBIOL-D-23-00748R1 

A mathematical, classical stratification modeling approach to disentangling the impact of weather on infectious diseases: a case study using spatio-temporally disaggregated Campylobacter surveillance data for  England and Wales

Dear Dr Lo Iacono,

I am pleased to inform you that your manuscript has been formally accepted for publication in PLOS Computational Biology. Your manuscript is now with our production department and you will be notified of the publication date in due course.

With kind regards,

Dorothy Lannert
